# Mesenchymal Stem Cells in the Treatment of COVID-19

**DOI:** 10.3390/ijms241914800

**Published:** 2023-09-30

**Authors:** Bei-Cyuan Guo, Kang-Hsi Wu, Chun-Yu Chen, Wen-Ya Lin, Yu-Jun Chang, Tai-An Lee, Mao-Jen Lin, Han-Ping Wu

**Affiliations:** 1Department of Pediatrics, National Cheng Kung University Hospital, College of Medicine, National Cheng Kung University, Tainan 70403, Taiwan; gbc628@gmail.com; 2Department of Pediatrics, Chung Shan Medical University Hospital, Taichung 40201, Taiwan; cshy1903@gmail.com; 3School of Medicine, Chung Shan Medical University, Taichung 40201, Taiwan; 4Department of Emergency Medicine, Tungs’ Taichung Metro Harbor Hospital, Taichung 43503, Taiwan; yoyo116984@gmail.com; 5Department of Nursing, Jen-Teh Junior College of Medicine, Nursing and Management, Miaoli 35664, Taiwan; 6Department of Pediatrics, Taichung Veterans General Hospital, Taichung 43503, Taiwan; 7Laboratory of Epidemiology and Biostastics, Changhua Christian Hospital, Changhua 50006, Taiwan; 83686@cch.org.tw; 8Department of Emergency Medicine, Chang Bing Show Chwan Memorial Hospital, Changhua 50544, Taiwan; d0i6a0n9e@hotmail.com; 9Division of Cardiology, Department of Medicine, Taichung Tzu Chi Hospital, The Buddhist Tzu Chi Medical Foundation, Taichung 42743, Taiwan; 10Department of Medicine, College of Medicine, Tzu Chi University, Hualien 97002, Taiwan; 11College of Medicine, Chang Gung University, Taoyuan 33302, Taiwan; 12Department of Pediatrics, Chiayi Chang Gung Memorial Hospital, Chiayi 61363, Taiwan

**Keywords:** COVID-19, mesenchymal stem cell, treatment

## Abstract

Since the emergence of the coronavirus disease 2019 (COVID-19) pandemic, many lives have been tragically lost to severe infections. The COVID-19 impact extends beyond the respiratory system, affecting various organs and functions. In severe cases, it can progress to acute respiratory distress syndrome (ARDS) and multi-organ failure, often fueled by an excessive immune response known as a cytokine storm. Mesenchymal stem cells (MSCs) have considerable potential because they can mitigate inflammation, modulate immune responses, and promote tissue regeneration. Accumulating evidence underscores the efficacy and safety of MSCs in treating severe COVID-19 and ARDS. Nonetheless, critical aspects, such as optimal routes of MSC administration, appropriate dosage, treatment intervals, management of extrapulmonary complications, and potential pediatric applications, warrant further exploration. These research avenues hold promise for enriching our understanding and refining the application of MSCs in confronting the multifaceted challenges posed by COVID-19.

## 1. Introduction

Since 2020, the relentless grip of the coronavirus disease 2019 (COVID-19) has profoundly shaped people’s lives worldwide, causing extensive global devastation [1,2]. As of 13 September 2023, 770,563,467 cases of COVID-19 have been confirmed and countless lives lost [3,4]. The COVID-19 virus (SARS-CoV-2) primarily spreads through large respiratory droplets, airborne particles, and fecal–oral and surface contact [5,6]. These modes of transmission significantly contribute to the rapid and difficult-to-control spread of the virus, despite the implementation of various government response policies in many countries. Consequently, the global community has actively dedicated itself to identifying effective methods for preventing and treating the disease. COVID-19 vaccines have been developed to prevent the spread of the disease and end the pandemic [7]. Significant progress has also been made in developing antiviral drugs, which have proven instrumental in treating the disease [8,9]. However, their effectiveness against COVID-19 has declined with the emergence of new virus variants [10]. COVID-19 can be categorized into mild, moderate, severe, and critical [11]. In mild and moderate COVID-19 cases, supportive care, oxygen therapy, and antiviral treatment are important for treatment [12,13]. However, in patients with severe and critical COVID-19, hyperinflammation and a cytokine storm resulting from a dysregulated host innate immune response often occurs [14,15], leading to the potential development of acute respiratory distress syndrome, septic shock, multiple organ damage, and even death [16,17,18,19,20]. In addition to the traditional supportive care for viral diseases, various strategies have been proposed to address the dysregulated immune response and mitigate the inflammatory cascade of COVID-19.

Immunotherapy-related agents, such as glucocorticoids and anti-interleukin-6 (IL-6) receptor antibodies, have effectively modulated the immune response and reduced excessive inflammation in these patients [21]. These treatments have demonstrated promising results in reducing mortality among individuals affected by the disease [22,23,24,25]. However, it is important to consider the potential risk of secondary infections associated with these therapies [26,27].

Unlike immunosuppressants, mesenchymal stem cells (MSCs) possess a unique combination of properties, including the ability to reduce inflammation, immunomodulation, and regenerative capabilities [28]. COVID-19 primarily affects the respiratory tract and can potentially lead to the development of acute respiratory distress syndrome (ARDS), cardiac injury, acute kidney injury, and multi-organ failure, often triggered by a cytokine storm [29,30]. MSC-based therapy has been proposed as a promising option for managing COVID-19 due to its ability to significantly suppress cytokine storms, leading to improved clinical symptoms and higher survival rates [31]. In this review, our objective is to investigate clinical trials related to MSCs in the treatment of COVID-19, as published in the scientific literature. We aim to provide a comprehensive summary that covers clinical symptoms, biomarkers associated with the cytokine storm, and lung imaging within the context of MSC therapy for COVID-19. Additionally, we will include a summary of the pathology and organ damage aspects. Furthermore, we will discuss potential future directions in this field.

## 2. Pathology and Vital Organ Damage in COVID-19

### 2.1. Mechanisms of Entry and Exit of COVID-19

The pathogenesis of COVID-19 is complex. The coronavirus virion comprises four structural proteins: envelope (E), membrane (M), nucleocapsid (N), and spike (S) [32]. The viral S protein binds to the ACE2 receptor via the receptor-binding domain (RBD) of the S1 subunit, facilitating viral entry into various human host cells, including those in the vascular endothelium, lungs, heart, brain, kidneys, intestines, liver, and pharynx [33,34,35,36]. Type II alveolar epithelial cells are considered to be the cells most predisposed to viral infections [37]. Upon RBD-receptor interaction, the S protein undergoes proteolytic cleavage by enzymes such as furin, or cellular proteases such as transmembrane serine protease 2 (TMPRSS2) in both the S1 and S2 subunits [33,38,39,40,41]. This cleavage exposes the fusion peptide of the S2 protein, enabling the S2 subunit trimers to fuse directly with the host cell membrane [42]. This process leads to deposition of the virus-positive RNA genome into the host cell. Another entry route that the virus may use is through the endosome. This occurs when the target cell expresses insufficient TMPRSS2 or if the virus-ACE2 complex does not encounter TMPRSS2. In such cases, the virus-ACE2 complex is internalized via clathrin-mediated endocytosis into endolysosomes, where S2’ cleavage is performed by cathepsins. Cathepsins require an acidic environment for their activity [33,43]. Furthermore, various membrane proteins can act as ACE2 cofactors or alternative receptors, enabling SARS-CoV-2 entry into different cell types, even those that do not typically express ACE2 [44]. However, their specific roles in SARS-CoV-2 pathogenesis remain unclear [45].

After entering the host cells, the viral genome hijacks the host cell machinery to initiate replication and translation. The molecular machinery of the host cell reads viral genetic material, leading to the synthesis of viral proteins. Among these proteins are the envelope glycoproteins processed within the Golgi apparatus [33]. When virus-containing vesicles fuse with the plasma membrane, assembled viruses are released into the extracellular space. This process occurs outside the cell.

### 2.2. Cytokine Storm and Severity of COVID-19

SARS-CoV-2 primarily infects type 2 pneumocytes that express ACE2 receptors in the alveoli and trigger an immune response characterized by the production of inflammatory cytokines and a weak interferon (IFN) response [46]. After cytokine release, membrane-bound immune receptors and downstream signaling pathways play crucial roles in mediating the proinflammatory immune responses of pathogenic Th1 cells and intermediate CD14+CD16+ monocytes. This process promotes the infiltration of macrophages and neutrophils into lung tissue, contributing to the cytokine storm’s onset [47]. Proinflammatory cytokines, including granulocyte-macrophage colony-stimulating factor (GM-CSF) and IL-6, are secreted by pathogenic Th1 cells. GM-CSF then activates CD14+CD16+ monocytes, leading them to produce significant quantities of IL-6, tumor necrosis factor-alpha (TNF-α), and other cytokines [48]. Neutrophil extracellular traps, which are extracellular nets released by neutrophils, can potentially contribute to the release of cytokines [49]. Cytokine storms are primarily characterized by the overexpression of IL-6 and TNF-α, two proinflammatory cytokines that play a significant role in the exaggerated immune response observed in severe cases of the disease.

In severe cases of COVID-19, multiple studies have indicated higher levels of IL-2, IL-6, IL-7, IL-10, CXCL10(IP-10), CCL2 (MCP-1), TNF-α, CCL3(MIP-1α), and G-CSF when compared to patients with mild and moderate infections [14,21,50,51,52,53]. Furthermore, increased levels of proinflammatory cytokines have been observed in patients with COVID-19 [54]. Patients experiencing severe COVID-19 demonstrate significant and sustained decreases in lymphocyte counts, particularly CD4+ and CD8+ T cells, but show increases in neutrophil counts when compared to those with milder cases [50,55]. The loss of T cells may worsen specific pathological inflammatory responses during SARS-CoV-2 infection [56], whereas the restoration of T cells could potentially alleviate inflammatory responses during SARS-CoV-2 infection [51]. Thus, T-cell lymphopenia, and dynamic cytokine storm are associated with COVID-19 severity [57,58]. Therefore, these findings highlight the need for clinicians to identify patients at risk of developing severe COVID-19 as early as possible through monitoring the dynamic cytokine storms.

## 3. Organ Damage of COVID-19

The severity of COVID-19 encompasses a broad spectrum, ranging from asymptomatic cases to mild presentations characterized by upper respiratory symptoms, such as fever, cough, sneezing, fatigue, and sore throat. This progression can lead to moderate illness with pneumonia, and in severe cases, individuals may experience dyspnea and hypoxemia, resulting in blood oxygen levels falling below the critical threshold of 94% [59,60]. The impact of the virus extends beyond the lungs and can damage other organs [61,62,63]. The pulmonary and extrapulmonary manifestations of COVID-19 are presented in Table 1. In more critical scenarios, COVID-19 can cause a range of severe conditions, including acute respiratory distress syndrome, shock, encephalopathy, heart failure, acute kidney injury, coagulopathy, and multiple organ dysfunction [60,64]. Cytokine storms are widely recognized as significant contributors to the development of multi-organ failure in COVID-19 [65].

### 3.1. Lung Involvement in COVID-19

SARS-CoV-2 primarily targets the lungs, and its severity ranges from asymptomatic to respiratory failure or death [66]. Inhaled viral particles of COVID-19 are primarily deposited in the nasal mucosa, where the virus infects and replicates within the epithelial target cells [67]. Infection leads to alveolar injury and inflammation in the lung tissue [68]. Dendritic cells (DCs) and alveolar macrophages phagocytose virus-infected apoptotic epithelial cells, initiating T-cell responses that activate both innate and adaptive immune mechanisms [69].

One critical aspect of COVID-19 pathology is the cytokine storm, characterized by an excessive release of proinflammatory cytokines and chemokines like TNF-α, IL-1β, and IL-6 [29]. This immune response contributes to immunopathology, leading to lung damage and, in some cases, immune suppression characterized by reduced T-cell counts and increased vulnerability to bacterial infections [70,71].

Autopsy studies of patients with COVID-19 pneumonia revealed acute interstitial pneumonia and diffuse alveolar damage. The lung tissue showed macrophage infiltration, hyaline membrane formation, and alveolar wall edema. Microvascular involvement includes hyaline thrombosis, hemorrhage, vessel wall edema, and immune cell infiltration [72]. Lung-restricted vascular immunopathology, also known as “diffuse pulmonary intravascular coagulopathy” (PIC), is associated with COVID-19, particularly in severe cases accompanied by ARDS [73].

An interesting feature of COVID-19 pneumonia is “silent hypoxia,” wherein patients tolerate low oxygen levels without dyspnea [74,75]. This unique characteristic distinguishes it from typical ARDS and is attributed to preserved lung compliance and high air volume with hypocapnia [75,76]. The disease presents with two time-associated phenotypes. Early disease stages show an L-phenotype with high lung compliance and a low Va/Q ratio. The lung weight and recruitment were low. Progression to the H phenotype includes decreased compliance, edema, and shunting. Type L patients can improve or worsen, transitioning to type H owing to COVID-19 pneumonia and high-stress ventilation [77,78].

### 3.2. Cardiovascular System Involvement in COVID-19

COVID-19 patients with hypertension, diabetes, and cardiovascular disease experience worse outcomes [79,80]. Cardiovascular complications, such as myocardial injury, heart failure, and arrhythmias are associated with poor survival [81,82]. Obesity is also a predictor of adverse cardiovascular outcomes in COVID-19 [83]. Myocardial injury is a notable finding in COVID-19 patients and is believed to be facilitated by elevated ACE2 expression in the cardiac tissue, allowing for direct virus-induced damage [84,85]. Isolated cases of COVID-19-induced myocarditis have been reported, suggesting that direct myocardial injury caused by the virus is a possible mechanism [86,87]. The binding of SARS-CoV-2 to ACE2 may lead to downregulation of ACE2, potentially compromising heart function [88]. COVID-19 can induce a hyper-inflammatory state with increased levels of inflammatory cytokines, rendering atherosclerotic lesions more vulnerable to disruption and increasing the risk of acute coronary syndrome [89,90]. The direct effect of the virus on endothelial cells and the host inflammatory response can lead to endothelial dysfunction in various organs, including the heart [91]. Myocardial injury can result from a mismatch between oxygen supply and demand, known as type 2 myocardial infarction [92]. Hypotension in sepsis and cytokine storms reduces myocardial perfusion [93], whereas infection and fever increase myocardial cell demand [94,95].

Fulminant myocarditis may occur as a manifestation of COVID-19 and lead to left ventricular dysfunction or cardiogenic shock [96]. Heart failure and arrhythmias are more prevalent in patients with severe COVID-19, particularly critically ill patients [97,98]. In addition, Kawasaki-like syndrome, characterized by circulatory dysfunction and macrophage activation, has been reported in some COVID-19 patients [99], suggesting a potential predisposing mechanism associated with the virus’s cytokine storm. 

### 3.3. Gastrointestinal Tract and Liver Involvement in COVID-19

Digestive symptoms in COVID-19 patients, including diarrhea, nausea, vomiting, and abdominal pain, are likely linked to the affinity of the virus for ACE2 receptors found abundantly in the GI tract, including the intestine, liver, and pancreas [100,101]. These symptoms can be the primary manifestations in COVID-19 patients, potentially leading to delayed diagnosis owing to their nonspecific nature [102,103]. SARS-CoV-2 shedding has been detected in the stool of COVID-19 patients, and as with most enteric viruses, it could be a potential fecal–oral transmission route [104]. 

Mild and transient liver injury, as well as severe liver damage, can occur during COVID-19 [105]. However, the mechanism of liver injury is not fully understood and can be caused by direct viral infection of hepatocytes, immune-related injury, or drug hepatotoxicity [106]. A systematic review and meta-analysis found that patients with severe COVID-19 have a higher risk of developing gastrointestinal symptoms and liver injury than those with non-severe disease [107]. 

### 3.4. Hematological Involvement in COVID-19

COVID-19 is a systemic infection that significantly affects the hematopoietic system and homeostasis. The cytokine storm is likely a key factor underlying the observed lymphopenia [108]. Lymphopenia is more frequent in moderate and severe COVID-19 cases, making it a valid marker of disease severity and mortality [108,109]. Neutrophilia has been identified as a predictor of poor outcomes in COVID-19 because severe disease is linked to an increased neutrophil-to-lymphocyte ratio and elevated expression of the neutrophil-related cytokines IL-8 and IL-6 in the serum [110,111]. COVID-19-induced severe lung inflammation may impair platelet production in the lungs because the lungs possess potential hematopoietic functions and serve as a primary site for terminal platelet production, accounting for approximately half of the total platelet production [110,112]. Thrombocytopenia is associated with an increased risk of severe disease and mortality in patients with COVID-19 [113,114]. Therefore, it should be considered a crucial clinical indicator of worsening illness during hospitalization [113]. Elevated D-dimer levels and coagulation abnormalities indicate blood hypercoagulability, particularly in severe COVID-19 cases [115]. Biomarkers such as high serum procalcitonin, C-reactive protein (CRP), and ferritin levels at presentation have emerged as poor prognostic factors associated with severe disease and poor survival [116]. COVID-19 is associated with a high risk of venous thromboembolism [117], and prompt thromboprophylaxis with low-molecular-weight heparin is recommended [118].

### 3.5. Neurological Involvement in COVID-19

SARS-CoV-2 can infiltrate the nervous system through two main routes: directly by utilizing the olfactory nerve pathway or by compromising the blood–brain barrier (BBB); and indirectly by triggering a vigorous immune response resulting in systemic inflammation or hypoxia [119]. COVID-19 can lead to neurological symptoms affecting the central nervous system (CNS), peripheral nervous system (PNS), and skeletal muscles in approximately one third of patients [120,121]. Common symptoms include dizziness, headache, taste impairment, anosmia, impaired consciousness, and nerve pain [122]. Severe cases of COVID-19 have a higher prevalence of neurological symptoms than milder cases [123]. Encephalopathy is a devastating CNS complication frequently encountered in older and critically ill patients. It arises from hypoxic/metabolic changes caused by cytokine storms and has severe consequences [124]. Moreover, patients with severe disease or individuals with vascular risk factors, such as hypertension, diabetes mellitus, or dyslipidemia, are at an increased risk of acute cerebrovascular diseases, primarily acute ischemic stroke [125]. Additionally, patients with stroke and concurrent COVID-19 were found to have a significantly higher risk of severe disability and death than those without COVID-19 [126].

### 3.6. Kidney Involvement in COVID-19

In COVID-19, kidney-related symptoms such as proteinuria and dipstick hematuria are not uncommon [127,128] and have been identified as significant predictors of severe or critically severe cases [127].

Acute kidney injury (AKI) affects approximately 30% of hospitalized COVID-19 patients and is one of the most frequent extrapulmonary complications [129]. Kidney damage induced by SARS-CoV-2 is believed to be multifactorial. ACE2, a receptor in various kidney cells, plays a significant role in COVID-19-related kidney abnormalities [130]. Other factors, including macrophage activation syndrome, cytokine storm, lymphopenia, endothelial dysfunction, organ interactions, hypercoagulability, rhabdomyolysis, and sepsis, can also contribute to AKI development [131]. The rate of AKI is higher in severe COVID-19 cases [132,133], and patients in the intensive care unit (ICU) show elevated levels of inflammatory cytokines, suggesting a potential association with cytokine release syndrome (CRS) leading to kidney injury [53].

### 3.7. Endocrine Involvement in COVID-19

Patients with diabetes mellitus or obesity have an increased risk of developing severe manifestations of COVID-19 [83,134]. The infection caused by SARS-CoV-2 can trigger a hyperglycemic state and potentially lead to ketoacidosis in both diabetic and non-diabetic patients, resulting in critical outcomes [135,136]. Notably, ACE2 expression has been identified in the endocrine pancreas [137], suggesting that SARS-CoV-2 directly binds to ACE2 on-cells, potentially contributing to insulin deficiency and hyperglycemia [138]. SARS-CoV-2-related factors, including increased cytokine levels, may adversely affect pancreatic β-cell function and induce apoptosis. This can result in diminished insulin production and the onset of ketosis [139]. Well-managed diabetic condition has been correlated with a significantly reduced risk of mortality and the need for invasive mechanical ventilation in individuals with COVID-19 [140].

## 4. MSCs in COVID-19 Treatment

### 4.1. Source and Immunomodulation of MSCs

MSCs, a type of versatile stem cells, can be sourced from various tissues of both adult and neonatal origin. These tissues include the bone marrow (BM), adipose tissue (AT), and peripheral blood [91] from adults, as well as neonatal birth-associated tissues, such as the umbilical cord (UC), cord blood (CB), and placenta (PL) [141]. Derived from the ethereal realm of fetal tissues, particularly the umbilical cord, MSCs have emerged as a true spectacle and have the advantage of rapid proliferation and enhanced immunomodulatory properties. This sets them apart from other sources, sidestepping any inconveniences that might arise while procuring MSCs from the bone marrow or adipose tissue [142]. 

MSCs possess differentiative and regenerative attributes and can secrete factors, such as hepatocyte growth factor, vascular endothelial growth factor, and keratinocyte growth factor. These molecules play crucial roles in regenerating type II alveolar epithelial cells [143]. Moreover, MSCs can be drawn to sites of inflammation through diverse chemokines and can regulate the activities of various immune cells, including NK cells, dendritic cells, B cells, T cells, neutrophils, and macrophages. This modulation occurs via direct and paracrine mechanisms. The major effectors of this process include indoleamine 2,3-dioxygenase, transforming growth factor β, human leukocyte antigen isoforms, and prostaglandin E2 [144]. Thus, MSCs possess immunomodulatory properties, can reduce immunity, and may offer a therapeutic option for patients with severe or critical COVID-19 to suppress overactivated inflammatory responses. 

### 4.2. Effectiveness of MSC Treatment in Other Diseases

MSCs have demonstrated significant clinical efficacy in the treatment of various immune disorders. In animal studies, human umbilical cord-derived MSCs have shown promising results in alleviating cytokine storms and reducing lung damage in mice with LPS-induced acute lung injury (ALI) [145]. Furthermore, a review highlighted the positive impact of MSC therapy on survival rates, hematopoietic reconstitution, and the recovery of peripheral blood cells in animal models of aplastic anemia [146]. In clinical practice, MSCs have been successfully used to treat numerous immune disorders [147], including inflammatory bowel disease [148,149,150], systemic lupus erythematosus [151,152,153], graft-versus-host disease [154,155], or multiple sclerosis [156,157,158]. MSCs exert significant effects on tumor growth and treatment response, making them promising candidates for cancer treatment, either alone or in combination with other therapies [159,160]. MSCs have the potential to effectively treat severe COVID-19 and its complications. 

### 4.3. Current Outcomes of Clinical Trials of MSCs in COVID-19 Treatment

Numerous clinical trials involving MSCs have demonstrated their effectiveness in treating COVID-19 and its associated complications. Currently, over 100 registered clinical trials have investigated the use of MSCs for COVID-19 treatment [161]. Most of these trials focused on severely ill COVID-19 patients. Nevertheless, a phase I clinical study has provided evidence of the safety of both high and low doses of DW-MSC infusion in patients with non-severe COVID-19 [162]. To substantiate the therapeutic efficacy of MSCs in these patients, large-scale randomized controlled trials could be imperative for definitive confirmation. Our review encompassed more than 20 clinical trials, the details of which are presented in Table 2. It is important to note that all the aforementioned clinical trials have demonstrated the safety of MSC treatment. The effectiveness of the MSC treatment in terms of clinical symptoms, biomarkers, and medical imaging in clinical trials is discussed below, with summary provided in Table 3.

#### 4.3.1. Effectiveness of MSC Treatment in Clinical Symptoms

MSC therapy has shown promising outcomes in reducing clinical symptoms in patients with severe COVID-19. As stated by Shu et al. [163], individuals who underwent treatment with umbilical cord mesenchymal stem cells (UC-MSC) experienced enhancements in clinical symptoms, such as weakness, fatigue, shortness of breath, and improved oxygenation index as early as the third day of treatment. Prenatal MSCs derived from umbilical cord (UC-MSC) or placental (PL-MSC) tissues can be utilized to treat critically ill patients with COVID-19-induced ARDS, resulting in reduced dyspnea and increased SpO2 within 2–4 days after the initial infusion in 64% of patients [164]. A multicenter randomized, double-blind trial unveiled a significant increase in PaO2/FiO2 ratios in the UC-MSC group compared to the placebo group in COVID-19-associated ARDS [165]. In a clinical trial using human menstrual blood-derived mesenchymal stromal cells for the treatment of severe and critically ill COVID-19 patients, significant improvement in dyspnea after MSC infusion on days 1, 3, and 5 and significant improvements in SpO2 and PaO2 following MSC infusion were noted. Additionally, patients in the MSC group showed significantly lower mortality (7.69% in the experimental group vs. 33.33% in the control group; *p* = 0.048) [166]. Farkhad et al. demonstrated that mesenchymal stromal cell therapy could improve the SPO2/FIO2 ratio in COVID-19-induced ARDS patients [167]. Sadeghi et al. showed that in COVID-19-induced ARDS cases treated with placenta-derived decidual stromal cells, there was a noticeable improvement in oxygenation levels, with a median increase from 80.5% (range, 69–88) to 95% (range, 78–99) (*p* = 0.012) [168].

Lanzoni et al. [169] reported improved patient survival and a shorter time to recovery after administering two rounds of intravenous allogeneic UC-MSC to patients with ARDS. In a phase I/II study involving patients with severe COVID-19, survival rates were notably higher in the MSC group at both 28 and 60 days after BM-MSC treatment (both *p* values < 0.05) [170]. In Indonesia, a randomized controlled trial demonstrated a survival rate 2.5 times higher in the UC-MSC group than in the control group in critical COVID-19 cases [171]. Fathi-Kazerooni et al. demonstrated that the survival rate in severe COVID-19 patients treated with human Mesenchymal Stromal Cells was significantly higher than that of patients who received placebo treatment (*p* < 0.001) [172].

Following a 2-year monitoring period for severe COVID-19 patients subjected to UC-MSC treatment, a slightly smaller subset of individuals within the MSC-treated group exhibited a 6 min walking distance (6-MWD) falling below the lower boundary of the normal range in comparison to the placebo-administered group (OR = 0.19, 95% CI: 0.04–0.80, Fisher’s exact test, *p* = 0.015). Furthermore, at month 18, the MSC-treated group demonstrated a higher general health score on the Short Form 36 questionnaire than the placebo group (MSC group: 50.00, placebo group: 35.00) (95% CI: 0.00–20.00, Wilcoxon rank sum test, *p* = 0.018) [173].

#### 4.3.2. Effectiveness of MSC Treatment in Biomarkers Related to Cytokine Storm

Many inflammatory biomarkers and cytokines are closely associated with severe and critical COVID-19. Notably, significant improvements in the levels of these biomarkers were observed after MSC treatment. Shu et al. [163] reported that CRP and IL-6 levels significantly decreased starting from day three, and the time required for lymphocyte counts to return to the normal range was notably shorter in patients with severe COVID-19 following UC-MSC infusion. In a study by Lanzoni et al. [169], significantly lower concentrations of GM-CSF, IFN-r, IL-5, IL-6, IL-7, TNF-a, and TNF-b were observed in COVID-19-related acute respiratory distress syndrome. Meng et al. [174] showed an improvement in the percentage of inspired oxygen (PaO2/FiO2) ratio and a declining trend in levels of inflammatory cytokines, including high levels of IL-6, IFN-γ, TNF-α, MCP-1, interferon-inducible cytokine IP-10 (IP-10), IL-22, interleukin 1 receptor type 1 (IL-1RA), IL-18, IL-8, and macrophage inflammatory protein 1-alpha (MIP-1), after three rounds of UC-MSC treatment. Fathi-Kazerooni et al. conducted a study that showed that in severe COVID-19 patients treated with human mesenchymal stromal cells, the levels of CRP on day five were notably lower than those in patients who underwent placebo treatment (*p* < 0.03). Additionally, within the treatment group of critical COVID-19 patients, significant reductions in CRP, LDH, D-dimer, and ferritin levels were observed after MSC treatment (all *p* < 0.05) [172]. Sadeghi et al. demonstrated that in cases of COVID-19-induced ARDS treated with placenta-derived decidua stromal cells, there were significant reductions in levels of IL-6, with a median decrease from 69.3 (range 35.0–253.4) to 11 (range 4.0–38.3) pg/mL, and CRP, with a median decrease from 69 (range 5–169) to 6 (range 2–31) mg/mL (*p* = 0.028) [168]. After aerosol inhalation of exosomes derived from human adipose-derived MSCs (haMSC-Exos) in COVID-19 patients, a pilot study reported an increase in lymphocyte counts and a decrease in CRP, LDH, and IL-6 levels [175].

In a case series study, considerable reductions in the serum levels of TNF-α, IL-8, and CRP were evident in all six critically ill COVID-19-induced ARDS survivors [164]. In a randomized clinical trial involving critically ill COVID-19 patients who underwent mesenchymal stromal cell infusion, the UC-MSC group exhibited reduced ferritin, IL-6, and MCP1-CCL2 levels on the fourteenth day. In the second month, decreases in reactive C-protein, D-dimer, and neutrophil levels were observed, along with increases in the numbers of TCD3, TCD4, and NK lymphocytes [176]. In a phase I/II clinical trial, patients with severe COVID-19 treated with BM-MSCs exhibited significantly lower D-dimer levels on day seven than control patients [170]. 

An Indonesian randomized controlled trial revealed that the infusion of UC-MSC led to a significant decrease in IL-6 levels in recovered COVID-19 patients with ARDS (*p* = 0.023) [171]. A randomized controlled trial showed that MSC infusion was associated with a decrease in inflammatory cytokines, such as IL-6 (*p* = 0.015), TNF-α (*p* = 0.034), IFN-γ (*p* = 0.024), and CRP (*p* = 0.041) in ARDS COVID-19 patient [177]. A prospective double-controlled trial revealed that after adding MSCs transplantation therapy to treat critical COVID-19 patients in the ICU, serum ferritin, fibrinogen, and CRP levels significantly decreased compared to conventional therapy alone [178]. In a successful phase 1 clinical trial with a control-placebo group, a noteworthy reduction (*p* < 0.05) was noted in biomarkers such as CRP, IL-6, IFN-γ, TNF-α, and IL-17A in COVID-19-induced ARDS patients. Conversely, there was a significant increase in the serum levels of TGF-B, IL-1B, and IL-10 [167]. In a phase 1 clinical trial, a generalized estimating equation analysis showed a significant decrease in ferritin levels (*p* = 0.008) after MSC treatment in COVID-19 patients [179].

#### 4.3.3. Effectiveness of MSC Treatment in Lung Image

Among COVID-19 patients, the most frequently observed computed tomography (CT) findings include ground-glass opacification, infiltration, consolidation, pneumonia, and emphysematous changes [185]. After MSC treatment in COVID-19 patients, many clinical trials have shown improved lung change. Shu et al. [163] reported shorter lung inflammation absorption on CT imaging in patients with severe COVID-19 in the UC-MSC group than in the control group. In a clinical trial involving a case series, lung CT revealed remarkable indications for recovery, including a reduction in ground-glass opacities or consolidations, among patients with COVID-19-induced ARDS [164]. Chest CT images showed complete fading of lung lesions within 2 weeks in moderate COVID-19 patients after UC-MSCs transfusion in the study by Meng et al. [174]. In a randomized, double-blind, placebo-controlled phase 2 trial, UC-MSCs led to a significant reduction in the proportions of solid component lesion volume compared to the placebo group in patients with COVID-19-induced ARDS (*p* = 0.043) [180]. A decrease in the extent of lung damage was observed in the fourth month after three rounds mesenchymal stromal cell infusion in critical COVID-19 patients in a randomized clinical trial [176]. Different degrees of resolution of pulmonary lesions after aerosol inhalation of haMSC-Exos were observed in patients with COVID-19 in a pilot study [175]. Xu et al. demonstrated a noteworthy distinction between experimental and control groups concerning the improvement rate of chest CT findings during the initial month following MSC infusion in individuals with severe and critical COVID-19 [166]. Fathi-Kazerooni et al. showed that in severe COVID-19 patients treated with human mesenchymal stromal cells, the percentage of pulmonary involvement exhibited a significant improvement in the group receiving secretome treatment (*p* < 0.0001) [172]. Sadeghi et al. demonstrated that all pulmonary infiltrates disappeared in patients with COVID-19-induced ARDS treated with placenta-derived decidual stromal cells [168]. 

## 5. Future Prospect of MSC Treatment in COVID-19 Patients

MSCs are administered via multiple routes. The intravenous delivery of MSCs could potentially lead to their entrapment in the pulmonary microcirculatory system, allowing for selective homing to injured lobes in the lungs as well as their distribution to extrapulmonary organs [62,186,187,188]. Other administration routes, including intratracheal, intranasal, subcutaneous, and pulmonary artery, have also been explored for delivering stem cells to treat COVID-19 [189]. These local delivery offers several advantages, including prolonged cell half-life, increased utilization efficiency, and reduced off-target effects on other organs [189]. Intratracheal administration of MSCs has been shown to increase cell concentrations in the lungs and demonstrated efficacy in preclinical models of respiratory diseases [190,191]. However, it is important to consider that this route of administration may also increase the risk of virus spread in the case of COVID-19 [192]. Therapeutic intravenous doses of MSC per treatment ranged from 5 × 10^5^ cells/kg body weight to 2 × 10^8^ cells/dose from different studies (Table 2). A clinical trial has demonstrated that the dose of MSCs for the treatment of COVID-19 can be as high as 8 × 10^8^ cells/kg/dose (clinical trial number: NCT04366323). Given the dose-dependent nature of the immunosuppressive effects and the documented safety of various cell doses, a higher MSC dosage may be warranted for critically ill patients [142]. However, the implementation of higher dosages requires meticulous clinical trial evaluation to determine the potential risks and benefits. For COVID-19, two to three courses of MSC therapy may be appropriate, with additional courses if insufficient recovery is observed [164,169,174,180,193]. Previous studies have proposed a 3-day interval between each course of MSC treatment during the acute phase [169,174,180]. However, the optimal cell dose, treatment duration, and interval protocols for MSC administration in COVID-19 management remain unclear. It is essential to conduct thorough research and develop guidelines to address these gaps and uncertainties.

The efficacy of MSCs in treating COVID-19-related conditions, including ARDS and severe cases in adults, has been well documented. However, there remains a need for further investigation into the potential of MSCs to address extrapulmonary complications, such as cardiovascular dysfunction and kidney injury in COVID-19 patients. Additionally, exploring their application in treating pediatric cases is crucial to expand our understanding of their therapeutic utility, underscoring the potential of MSCs in managing bronchopulmonary dysplasia in newborns and suggesting that MSCs hold promise in reducing lung inflammation, enhancing pulmonary architecture, mitigating pulmonary fibrosis, and ultimately improving survival rates [194]. Another study demonstrated encouraging results in managing acute graft-versus-host disease in pediatric patients, illustrating the versatility of MSCs in pediatric care [195]. In a phase I/II randomized, placebo-controlled clinical trial, intravenous transplantation of MSCs significantly improved HbA1c and C-peptide levels in pediatric patients. Furthermore, this treatment modulates the balance between proinflammatory and anti-inflammatory cytokines, highlighting its potential therapeutic impact [196]. Future research efforts should prioritize the evaluation of the effects of MSC-based cell therapy in COVID-19 patients, particularly in terms of restoring organ functionality and reducing mortality rates within this subgroup. Additionally, there is a need to investigate the efficacy of MSC-based therapy in pediatric patients with severe COVID-19.

By pursuing these avenues, we can deepen our understanding of the diverse therapeutic capacities of MSCs in various aspects of COVID-19 management. This endeavor holds significant promise for advancing our ability to effectively address the multifaceted challenges posed by the disease.

## 6. Conclusions

MSCs demonstrated significant therapeutic potential owing to their ability to attenuate inflammation, modulate immune responses, and facilitate tissue regeneration. MSCs have the potential to effectively treat severe COVID-19 and its complications. In our review, we conducted extensive research by reviewing dozens of clinical studies related to MSC treatment for COVID-19. We categorized these studies into clinical symptoms, cytokine storm biomarkers, and lung imaging for discussion. Additionally, we explored the prospects of MSC treatment in this context. These clinical trials have generated evidence showcasing the effectiveness of MSCs in addressing various facets of severe COVID-19 and COVID-19-related ARDS. These trials have yielded positive outcomes by improving clinical symptoms, enhancing survival rates, optimizing oxygen saturation levels, influencing cytokine storm-related biomarkers, and positively affecting lung imaging results. In the future, there may be a need to delve deeper into several areas. Investigating optimal administration routes, determining appropriate cell dosages, defining treatment intervals for MSC therapy, understanding the impact of severe extrapulmonary damage treatment, and exploring the potential applications of MSCs in pediatric cases are important aspects that warrant further investigation. These research directions hold promise for refining our understanding and utilization of MSCs as a therapeutic approach against the multifaceted challenges posed by COVID-19.

## Figures and Tables

**Table 1 ijms-24-14800-t001:** Systemic manifestations of COVID-19 infection.

Organ System	Manifestations of COVID-19 Infection
Pulmonary involvement	Poor ventilation
	Acute respiratory distress syndrome (ARDS)
Cardiovascular involvement	Cardiovascular disorder
	Arrythmia
	Myocarditis
	Myocardial infarction
	Acute coronary syndrome (ACS)
	Acute cor pulmonale
	Kawasaki-like syndrome
Gastrointestinal and hepatic involvement	Nausea and vomiting
	Diarrhea
	Abdominal pain
	Anorexia
	Elevated bilirubin
	Elevated Aminotransferases
Renal involvement	Acute kidney injury
	Hematuria
	Proteinuria
Hematological involvement	Neutrophilia
	Lymphopenia
	Thrombocytopenia
	Elevated procalcitonin, CRP and ferritin levels
	Elevated d-dimer
	Deep venous thrombosis
	Pulmonary embolism
	Catheter-related thrombosis
Neurological involvement	Headache
	Dizziness
	Anosmia
	Ageusia
	Nerve pain
	Myalgia
	Increased depression
	Cerebral hemorrhage
	Stroke
	Acute necrotizing encephalopathy
	Guillain barre syndrome
Endocrine involvement	Hyperglycemia
	Diabetic ketoacidosis
Dermatological involvement	Petechiae
	Urticaria
	Vesicles

**Table 2 ijms-24-14800-t002:** Clinical trial on MSC infusion for patients with COVID-19.

No	Study Title	Trial ID NO	Phase	Indications	Source of MSCs	Route and Time of Administration	Dose of MSCs	Number of Patients	Country	Reference
1	Safety of DW-MSC infusion in patients with low clinical risk COVID-19 infection: a randomized, double-blind, placebo-controlled trial	NCT04535856	Phase 1	Low clinical risk COVID-19	Allogenic UC-MSC	Intravenous infusions, 1 round	High dose: 1 × 10^8^ cells/round or Low dose: 5 × 10^7^ cells/round	9	Indonesia	[162]
2	Treatment of severe COVID-19 with human umbilical cord mesenchymal stem cells	ChiCTR2000031494	Phase 1	Severe/critical COVID-19	Human umbilical cord	Intravenous administration 1 round	2 × 10^6^ cells/kg	41	China	[163]
3	Mesenchymal stem cells derived from perinatal tissues for treatment of critically ill COVID-19-induced ARDS patients: a case series	IRCT20200217046526N2	Phase 1	ARDS in COVID-19	Allogeneic umbilical cord/placenta	Intravenous infusions, 3 round (at days 0, 2, and 4)	2 × 10^8^ cells/round	11	Iran	[164]
4	Treatment of COVID-19-associated ARDS with mesenchymal stromal cells: a multicenter randomized double-blind trial	NCT04333368	Phase 2	ARDS in COVID-19	Umbilical cord	Intravenous infusions, 3 round (at days 1, 3 ± 1, and 5 ± 1)	1 × 10^6^ cells/kg/round	47	French	[165]
5	Evaluation of the safety and efficacy of using human menstrual blood-derived mesenchymal stromal cells in treating severe and critically ill COVID-19 patients: An exploratory clinical trial	ChiCTR2000029606	Phase 1	Severe and critical COVID-19	Allogenic menstrual blood	Intravenous infusions, 3 round (1, 3, 7)	Total 9 × 10^7^ cells	44	China	[166]
6	Mesenchymal stromal cell therapy for COVID-19-induced ARDS patients: a successful phase 1, control-placebo group, clinical trial	IRCT20160809029275N1	Phase 1	ARDS in COVID-19	Allogenic Umbilical cord	Intravenous infusions, 3 round (1, 3, 5)	1 × 10^6^ cells/kg/round	20	Iran	[167]
7	Conquering the cytokine storm in COVID-19-induced ARDS using placenta-derived decidua stromal cells	IRCT2017010531786N1	Phase 1/2	ARDS in COVID-19	Placenta	Intravenous infusions, 1 or 2 rounds	1 × 10^6^ cells/kg/round	10	Iran	[168]
8	Umbilical cord mesenchymal stem cells for COVID-19 acute respiratory distress syndrome: A double-blind, phase 1/2a, randomized controlled trial	NCT04355728	phase 1/2a	ARDS in COVID-19	Allogeneic umbilical cord	Intravenous infusions, 2 round (at days 0 and 3)	10 ± 2 × 10^7^ cells/round	24	USA	[169]
9	Bone marrow-derived mesenchymal stromal cell therapy in severe COVID-19: preliminary results of a phase I/II clinical trial	NCT04445454	Phase 1/2	Severe COVID-19	Bone marrow	Intravenous infusions, 3 round (1, 4 (±1), 7 (±1))	1.5–3 × 10^6^ cells/kg/round	32	Belgium	[170]
10	Umbilical cord mesenchymal stromal cells as critical COVID-19 adjuvant therapy: A randomized controlled trial	NCT04457609	Phase 1	ARDS in COVID-19	Allogenic umbilical cord	Intravenous infusions, 1 round	1 × 10^6^ cells/kg/round	40	Indonesia	[171]
11	Safety and efficacy study of allogeneic human menstrual blood stromal cell secretome to treat severe COVID-19 patients: clinical trial phase I & II	IRCT20180619040147N6	Phase 1/2	Severe COVID-19	Allogeneic human menstrual blood	Intravenous infusions, 5 round (for 5 consecutive days)	5 mL MenSCs-derived secretome diluted in 100 mL of normalsaline	30	Iran	[172]
12	Human mesenchymal stem cell therapy in severe COVID-19 patients: 2-year follow-up results of a randomized, double- blind, placebo-controlled trial	NCT04288102	Phase 2	Severe COVID-19	Allogenic umbilical cord	Intravenous infusions, 3 round (at days 0, 3, and 6)	4 × 10^7^ cells/round	100	China	[173]
13	Human umbilical cord-derived mesenchymal stem cell therapy in patients with COVID-19: a phase 1 clinical trial	NCT04252118	Phase 1	Moderate and severe COVID-19	Allogeneic umbilical cord	Intravenous infusions, 3 round (at days 0, 3, and 6)	3 × 10^7^ cells/round	18	China	[174]
14	Nebulized exosomes derived from allogenic adipose tissue mesenchymal stromal cells in patients with severe COVID-19: a pilot study	NCT04276987.	Phase 2a	Severe COVID-19	Allogeneic adipose tissue	Inhalation 5 round (at days 1–5)	2 × 10^8^ particles/round (total 2 × 10^9^)	7	China	[175]
15	Safety and long-term improvement of mesenchymal stromal cell infusion in critically COVID-19 patients: a randomized clinical trial	U1111-1254-9819	Phase 1/2	Critical COVID-19	Allogeneic umbilical cord	Intravenous infusions, 3 round (at days 1, 3, and 5)	5 × 10^5^ cells/kg/round	17	Brazil	[176]
16	Allogenic mesenchymal stromal cells and their extracellular vesicles in COVID-19 induced ARDS: a randomized controlled trial	IRCT20200217046526N2	Phase 2	ARDS in COVID-19	Allogenic perinatal tissue	1st round: intravenous infusions 2nd round: intravenous infusion or inhalation	IV: 1 × 10^8^ cells/round Inhalation: 2 × 10^8^ cells/round	43	Iran	[177]
17	The systematic effect of mesenchymal stem cell therapy in critical COVID-19 patients: a prospective double controlled trial	NCT04392778	Phase 1/2	Critical COVID-19	Umbilical cord	Intravenous infusions, 3 round (at days 0, 3, and 6)	3 × 10^6^ cells/kg/round	30	Turkey	[178]
18	Cell therapy in patients with COVID-19 using Wharton’s jelly mesenchymal stem cells: a phase 1 clinical trial	IRCT20190717044241N2	Phase 1	Severe COVID-19	Umbilical cord	Intravenous infusions, 3 round (at days 0, 3, and 6)	1.5 × 10^8^ cells/round	5	Iran	[179]
19	Effect of human umbilical cord-derived mesenchymal stem cells on lung damage in severe COVID-19 patients: a randomized, double-blind, placebo-controlled phase 2 trial	NCT04288102	Phase 2	Severe COVID-19	Umbilical cord	Intravenous infusions, 3 round (at days 0, 3, and 6)	4 × 10^7^ cells/round	100	China	[180]
20	Safety and efficacy assessment of allogeneic human dental pulp stem cells to treat patients with severe COVID-19: structured summary of a study protocol for a randomized controlled trial (phase I/II)	NCT04336254	Phase 1/2	Severe COVID-19	Allogeneic dental pulp	Intravenous infusions, 3 round (at days 1, 4, and 7)	3 × 10^7^ cells/round	20	China	[181]
21	Human placenta-derived mesenchymal stem cells transplantation in patients with acute respiratory distress syndrome (ARDS) caused by COVID-19 (phase I clinical trial): safety profile assessment	IRCT20200621047859N4.	Phase 1	ARDS in COVID-19	Allogenic placenta	Intravenous infusions, 1 round	1 × 10^6^ cells/kg/round	20	Iran	[182]
22	Nebulization therapy with umbilical cord mesenchymal stem cell-derived exosomes for COVID-19 pneumonia	ChiCTR2000030261	Phase 1	Pneumonia in COVID-19	Umbilical cord	Nebulization, 1 round	1 × 10^6^ cells/kg/round	7	China	[183]
23	A randomized trial of mesenchymal stromal cells for moderate to severe acute respiratory distress syndrome from COVID-19	NCT04371393	Phase 3	ARDS in COVID-19	Allogenic bone marrow	Intravenous infusions, 2 round (two infusions during the first week, with the second infusion 4 days after the first infusion (±1 day)	2 × 10^6^ cells/kg/round	222	USA	[184]

**Table 3 ijms-24-14800-t003:** Effectiveness of MSC treatment in clinical trial for patients with COVID-19.

Study Title	Indications	Effectiveness of MSC Treatment			Reference
		Clinical Symptoms	Cytokine Storm Biomarkers	Lung Image (Chest CT)	
Treatment of severe COVID-19 with human umbilical cord mesenchymal stem cells	Severe/critical COVID-19	Improvement of weakness, fatigue, shortness of breath, and oxygenation index as early as the third day	↓ CRP and IL-6	Shorter lung inflammation absorption	[163]
Mesenchymal stem cells derived from perinatal tissues for treatment of critically ill COVID-19-induced ARDS patients: a case series	ARDS in COVID-19	Reduced dyspnea and increased SpO2 within 2–4 days	↓ TNF-α, IL-8, and CRP	Reduction in ground-glass opacities or consolidation	[164]
Treatment of COVID-19-associated ARDS with mesenchymal stromal cells: a multicenter randomized double-blind trial	ARDS in COVID-19	Significant increase in PaO2/FiO2 ratios			[165]
Evaluation of the safety and efficacy of using human menstrual blood-derived mesenchymal stromal cells in treating severe and critically ill COVID-19 patients: an exploratory clinical trial	Severe and critical COVID-19	Significant improvement in dyspnea on days 1, 3, and 5 and significant improvements in SpO2 and PaO2 Significantly lower mortality		Lung clearly	[166]
Mesenchymal stromal cell therapy for COVID-19-induced ARDS patients: a successful phase 1, control-placebo group, clinical trial	ARDS in COVID-19	Improve the SPO2/FIO2 ratio	↓ CRP, IL-6, IFN-γ, TNF-α, and IL-17A ↑ TGF-B, IL-1B, and IL-10		[167]
Conquering the cytokine storm in COVID-19-induced ARDS using placenta-derived decidua stromal cells	ARDS in COVID-19	Improvement in oxygenation levels, with a median increase from 80.5% to 95%	↓ IL-6 and CRP	Pulmonary infiltrates disappeared	[168]
Umbilical cord mesenchymal stem cells for COVID-19 acute respiratory distress syndrome: A double-blind, phase 1/2a, randomized controlled trial	ARDS in COVID-19	Improved patient survival and a shorter time to recovery	↓ GM-CSF, IFN-r, IL-5, IL-6, IL-7, TNF-a, and TNF-b		[169]
Bone marrow-derived mesenchymal stromal cell therapy in severe COVID-19: preliminary results of a phase I/II clinical trial	Severe COVID-19	Higher survival rate in the MSC group at both 28 and 60 days	↓ D-dimer		[170]
Umbilical cord mesenchymal stromal cells as critical COVID-19 adjuvant therapy: A randomized controlled trial	ARDS in COVID-19	Survival rate 2.5 times higher in the UC-MSC group than in the control group	↓ IL-6		[171]
Safety and efficacy study of allogeneic human menstrual blood stromal cells secretome to treat severe COVID-19 patients: clinical trial phase I & II	Severe COVID-19	Higher survival rate	↓ CRP, LDH, D-dimer, and ferritin levels	Improvement of lung involvement	[172]
Human mesenchymal stem cell therapy in severe COVID-19 patients: 2-year follow-up results of a randomized, double- blind, placebo-controlled trial	Severe COVID-19	Higher general health score on the Short Form 36 questionnaire			[173]
Human umbilical cord-derived mesenchymal stem cell therapy in patients with COVID-19: a phase 1 clinical trial	Moderate and severe COVID-19		↓ IL-6, IFN-γ, TNF-α, MCP-1, IP-10, IL-22, IL-1RA, IL-18, IL-8, and MIP-1	Complete fading of lung lesions within 2 weeks	[174]
Nebulized exosomes derived from allogenic adipose tissue mesenchymal stromal cells in patients with severe COVID-19: a pilot study	Severe COVID-19		↓ CRP, LDH, and IL-6	Resolution of pulmonary lesions	[175]
Safety and long-term improvement of mesenchymal stromal cell infusion in critically COVID-19 patients: a randomized clinical trial	Critical COVID-19		↓ Ferritin, IL-6, and MCP1-CCL2, CRP, D-dimer, and neutrophil levels ↑ TCD3, TCD4, and NK lymphocytes	Decrease in the extent of lung damage was observed in the fourth month	[176]
Allogenic mesenchymal stromal cells and their extracellular vesicles in COVID-19 induced ARDS: a randomized controlled trial	ARDS in COVID-19		↓ IL-6, TNF-α, IFN-γ, and CRP		[177]
The systematic effect of mesenchymal stem cell therapy in critical COVID-19 patients: a prospective double controlled trial	Critical COVID-19		↓ Ferritin, fibrinogen, and CRP		[178]
Cell therapy in patients with COVID-19 using Wharton’s jelly mesenchymal stem cells: a phase 1 clinical trial	Severe COVID-19		↓ Ferritin		[179]
Effect of human umbilical cord-derived mesenchymal stem cells on lung damage in severe COVID-19 patients: a randomized, double-blind, placebo-controlled phase 2 trial	Severe COVID-19			Significant reduction in the proportions of solid component lesion volume	[180]

## Data Availability

Not applicable.

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
