# Peer review of "Mesenchymal Stem Cells in the Treatment of COVID-19"

_ijms, 2023, doi:10.3390/ijms241914800_

Round 1

Reviewer 1 Report

This is an interesting and well organized review on a very relevant topic, that is the use of MSCs as therapeutic agents against Covid-19. However, MSCs are studied also for their promising anti-cancer properties, therefore I would add one or two sentences on this important topic just for completion.   

Reviewer 2 Report

Although the review is very interesting, mostly because the authors expose a very new approach in the treatment of acute SARS-CoV-2 acute infection, I think that it is not adequate for publication in the actual manuscript version because diverse reasons, mostly formal ones.

The mission of a review is to offer a global and clear vision of the issue that has been reviewed after an analytical study of the bibliography, not simply to give all the existing information. And, it has not been the case of the present work. The authors give a lot of information but, without a clear structure facilitating the understanding of all the offered information to obtain the desirable clear global vision. To obtain this global vision it is necessary to know the quality of the existing bibliography as well as, in this work, the pros and cons of the analyzed therapeutic approach. As demonstrative that the authors have not done it is table 2. In this table, the authors just express the existing bibliography as a simple list of publications. The order and structure of the table have not any clear structure to ease the understanding. The information is also very scarce expressing the main result of efficacy of each study as well as secondary effects. It is clear that the authors do not offer any data for the reader to evaluate the quality of the obtained evidence. It is also notorious the absence of statistical data.

Another noteworthy point is the extension dedicated by the authors to explain the extense forms of body affectation related to COVID-19. I think that all this information does not add any value to understand the studied issue. It would be more interesting to explain general aspects of MSC taking into account the complexity of the procedure as well as the different technical options to do it. Technical differences that could influence clinical results. It is a complex issue, at least for clinicians. Because of this complexity, I think that the authors have to explain general aspects of the mesenchymal stem cell infusion as potential benefits in the treatment based on pathophysiological mechanisms and previous experiences in diseases other than COVID-19. They should also explain different procedures of infusion as well as the different existing procedures to obtain MSC, focusing in the potential effect of such differences in the efficacy and security when applied to clinical situations as COVID-19.

When considering the manuscript as a whole, it is quite difficult to understand and analyze the shown information to obtain a final conclusion. The authors should give the information in a structured way. For example, when describing in the text and table 2 the existing papers, the information should be structured with a logical criteria. For example, the studies could be ordered depending on the severity of COVID-19, or on the kind of study (Case reports, Phase I or II trial…) or if it has been in humans, in silico or animal models. In the same sense, the information offered in the text of the Results section needs to be exposed according to the same logical structure. For example, first describing the quality of the bibliography and characteristics of the studies, after that, clinical efficacy and secondary effects as well as limitants of the results. In the actual version the authors just describe studies and more studies and their results, but the is not a clear scheme or structure. 

From a methodological point of view, it is also necessary for this kind of studies that the authors explain the criteria applied to detect and choose the bibliography. It is also mandatory to establish/analyze the quality of the evidence to stablish the strength of the conclusions of the study. Related to this point, the authors express affirmations like MSCs have demonstrated significant therapeutic potential owing to their ability to attenuate inflammation, modulate immune responses, and facilitate tissue regeneration. MSCs have the potential to effectively treat severe COVID-19 and its complications”. On the contrary, in other part of the manuscript they affirm “To substantiate the therapeutic efficacy of MSCs in these patients, large-scale randomized controlled trials are imperative for definitive confirmation”, and “In the future, there will be a need to delve deeper into several areas. Investigating optimal administration routes, determining appropriate cell dosages, defining treatment intervals for MSC therapy”. The authors are saying that MSC have demonstrated to be useful, but at the same time they are saying that it is necessary to confirm. Another point I do not understand is when in the manuscript it is established that “understanding the impact of severe extrapulmonary damage treatment, and exploring the potential applications of MSCs in pediatric cases are important aspects that warrant further investigation”. Nowhere in the study has been established if the considered bibliography has been of studies made in pediatric, adult or in both kinds of populations. Because of this reason, I think that this affirmation as those previously indicated are speculative and not based on the evidence shown in the manuscript.

Because of all the aforementioned reasons, in my opinion the manuscript needs a deep general modification before being considered for publication. 

Reviewer 3 Report

The article is devoted to a topical subject and may be useful to IJMS readers. However, I have a few comments:

(148-150) The names of the cytokines are more correctly referred to as: CCL3 (MIP-1α), CCL2 (MCP-1), CXCL10 (IP-10), G-CSF.

(155) "CD4+ cells and CD8+ cells", more precisely CD4+ and CD8+ T cells.

(159-163) It is necessary to specify specific NLR values.

There are currently 332 review articles on the treatment of COVID-19 with mesenchymal stem cells in PubMed. Therefore, it is necessary to clarify the novelty of your research in the Introduction and Conclusion sections.

Reviewer 4 Report

In this paper, the authors have done a review about the use of mesenchymal stem cells (MSC) for the treatment of COVID-19. It is an interesting paper, I have only some suggestions to improve it:

- Major comments:

- The first part is a review of the pathophysiology of covid-19. It is a bit too long, and this would benefit from being summarized because the major findings are indicated in the table 1. 

- In contrast, the part concerning the MSC  could be more in-depth, notably table 2 (or make another table for greater readability); It could be of interest to indicate in this table (or a new one) the primary goals of the study and the results.

- Perhaps a more detailed paragraph to explain how, in general,  MSC have therapeutic efficiency could be of interest.

- Minor comments:

- In the introduction section, line 82, the authors indicate thatinterleukin-1 receptor antagonist (the abbreviation IL-R is not relevant and should be IL-1RA or IL-R indicated just after Interleukin-1 receptor) and the comment of the authors that theses medicines should not be routinely used in COVID-19 patients is an arbitrary comment of the authors, is not related to the topic of the paper, and should be deleted.

- Similarly, line 167, it is written that clinicians should monitot neutrophil and lymphocytes counts....and NLR. It is not related to the topic of the paper, and should be deleted. There is a lot of aper suggesting that other biologicaal markers could be more relevant in order to identify high risk COVID patients.
